# The Roles and Mechanisms of lncRNAs in Liver Fibrosis

**DOI:** 10.3390/ijms21041482

**Published:** 2020-02-21

**Authors:** Zhi He, Deying Yang, Xiaolan Fan, Mingwang Zhang, Yan Li, Xiaobin Gu, Mingyao Yang

**Affiliations:** 1College of Animal Science and Technology, Sichuan Agricultural University, Chengdu 611130, China; zhihe@sicau.edu.cn (Z.H.); xiaolanfan@sicau.edu.cn (X.F.); mwzhangkiz@hotmail.com (M.Z.); liyan@sicau.edu.cn (Y.L.); 2Farm Animal Genetic Resources Exploration and Innovation Key Laboratory of Sichuan Province, Sichuan Agricultural University, Chengdu 611130, China; 3College of Veterinary Medicine, Sichuan Agricultural University, Chengdu 611130, China; guxiaobin198225@126.com

**Keywords:** lncRNA, liver fibrosis, function, mechanism

## Abstract

Many studies have revealed that circulating long noncoding RNAs (lncRNAs) regulate gene and protein expression in the process of hepatic fibrosis. Liver fibrosis is a reversible wound healing response followed by excessive extracellular matrix accumulation. In the development of liver fibrosis, some lncRNAs regulate diverse cellular processes by acting as competing endogenous RNAs (ceRNAs) and binding proteins. Previous investigations demonstrated that overexpression of lncRNAs such as H19, maternally expressed gene 3 (MEG3), growth arrest-specific transcript 5 (GAS5), Gm5091, NR_002155.1, and HIF 1alpha-antisense RNA 1 (HIF1A-AS1) can inhibit the progression of liver fibrosis. Furthermore, the upregulation of several lncRNAs [e.g., nuclear paraspeckle assembly transcript 1 (NEAT1), hox transcript antisense RNA (Hotair), and liver-enriched fibrosis-associated lncRNA1 (lnc-LFAR1)] has been reported to promote liver fibrosis. This review will focus on the functions and mechanisms of lncRNAs, the lncRNA transcriptome profile of liver fibrosis, and the main lncRNAs involved in the signalling pathways that regulate hepatic fibrosis. This review provides insight into the screening of therapeutic and diagnostic markers of liver fibrosis.

## 1. Introduction

Liver fibrosis, a reversible wound healing response followed by excessive extracellular matrix (ECM) accumulation, is a major public health concern and is caused by increased synthesis and deposition of ECM components and decreased or unbalanced ECM degradation [1,2]. Furthermore, liver fibrosis is generated via continuous wound-healing reactions and liver damage caused by various chronic liver diseases, including non-alcoholic steatohepatitis (NASH), alcohol abuse, biliary obstruction, hepatitis B and C, and several other aetiologies [3,4] (Figure 1). Liver fibrosis may develop into a common final pathway of chronic hepatic diseases (e.g., cirrhosis and hepatocellular carcinoma), with a prevalence of 2–19% [4,5,6]. Then, if this pathology cannot be eradicated, liver fibrosis progresses to cirrhosis and eventually to liver failure and malignancy [7]. Thus, analysis of the aetiology of liver fibrosis is important to prevent and treat hepatic fibrosis-related diseases.

Hepatic stellate cells (HSCs) make up 5–8% of the total liver cells involved in maintaining growth, differentiation, and regeneration [8]. HSCs are the main cell type responsible for liver fibrosis. HSC activation plays a key role in liver fibrosis and mainly leads to the excessive accumulation of smooth muscle α-actin (α-SMA) expression, type I collagen, and ECM proteins in the liver [9,10]. HSCs are quiescent and store abundant vitamin A in normal hepatic tissue [11,12]. In contrast, quiescent HSCs lose their vitamin A stores and transdifferentiate into fibrogenic myofibroblast-like cells following hepatic injury of any aetiology [12,13,14]. Thus, the suppression of HSC activation is considered a potential treatment target for hepatic fibrosis.

Recently, available evidence has demonstrated that noncoding RNAs (ncRNAs) are crucial regulators for controlling gene expression and transducing cellular signals [15]. NcRNAs have been revealed as the key determinants of gene expression in the course of carcinogenesis and liver fibrogenesis [7,16,17,18]. Growing proof hints at a significant role for ncRNAs in liver fibrosis [19,20]. Thus, the potential of ncRNAs as targets and biomarkers for the diagnosis and treatment of liver fibrosis has been suggested. Long noncoding RNAs (lncRNAs) are a noncoding RNA family with a transcript length of more than 200 nucleotides (nts). LncRNAs play important roles in regulating liver-related fibrosis and disease [8,21,22]. To date, experimental data from previous reports has demonstrated that lncRNAs are involved in multiple signalling pathways, including hedgehog (Hh) [23], notch [24], p53 [25], and transforming growth factor beta1 (TGF-β1)/mothers against decapentaplegic protein 3 (smad3) [26], and that they can inhibit or promote liver fibrosis by suppressing or activating HSCs; these lncRNAs include hepatocellular carcinoma upregulated lncRNA (HULC) [27], metastasis-associated lung adenocarcinoma transcript 1 (MALAT1) [20], Hotair [22], HOXA distal transcript antisense RNA (HOTTIP) [28], and MEG3 [29]. In this review, we have summarized the roles and mechanism of lncRNAs in liver fibrosis and provid some references for future research on liver disease and hepatic fibrosis.

## 2. Function and Mechanism of LncRNAs

The classification, functions, and mechanisms of lncRNAs in biological organisms are very complex. Based on their genomic organization relative to protein-coding genes, lncRNAs are classified into six groups: sense/antisense exonic lncRNA, sense/antisense intronic lncRNA, intergenic lncRNA, and bidirectional lncRNA (Figure 2A) [30]. LncRNAs are involved in multiple pathological processes and biological functions such as malignancy, differentiation, apoptosis, development, survival, and proliferation [31]. LncRNAs can impact biological pathways and cellular activities by different mechanisms; for example, they can target transcription factors, block transcription of nearby genes, direct methylation complexes, and initiate chromatin remodelling [32,33] (Figure 2B). During the progression of liver fibrosis, a number of lncRNAs play roles in the promotion or suppression of fibrosis through binding of ceRNAs to miRNAs and directly binding proteins [34,35,36]. These processes will be introduced in detail in the subsequent articles of this series.

## 3. Analysis of Liver Fibrosis by LncRNA Transcriptome Profiles

LncRNA transcriptome profiles based on genome-wide and high-throughput screening methods provide comprehensive gene expression profiles of pathological conditions. LncRNAs may provide a deep and novel understanding of the mechanisms of liver fibrosis [37]. Identifying and characterizing the specific mechanisms by which lncRNAs contribute to the course of disease are crucial for the progress of new therapies to reverse, halt, or delay hepatic fibrosis. To date, extensive data on the lncRNA expression patterns associated with liver fibrosis have been obtained based on genome-wide screening of tissues, cell types, and HSCs from human and animal models (mice and rats).

### 3.1. Many LncRNAs Are Expressed in Human Hepatic Fibrosis Tissues and Cells

LncRNA expression data (more than 3600 lncRNAs) were specifically recognized in HSCs and under the conditions of TGF-βsignalling when compared to 43 human tissues and cell types [37]. Nearly 40% of those lncRNAs are specifically expressed, and more than 400 lncRNAs are uniquely enriched in HSCs compared to normal tissues and cell types [37]. A total of 139 lncRNA loci were induced and 242 lncRNA loci were found to be repressed by TGF-β signalling activation [37]. Furthermore, the first insights into lncRNA expression pattern changes in activated human HSCs have been reported [38]. A total of 34,146 differentially expressed (DE) lncRNAs were described during human HSC quiescence and activation, and the expression changes of only 1685 DE mRNAs and 3763 DE lncRNAs were statistically significant when activated hHSCs were compared to quiescent hHSCs [38]. Moreover, the lncRNA expression patterns of fibrotic and normal liver samples in NASH patients have been investigated [27]. In NASH patients, there are strong links between fibrosis and both liver-related mortality and a heightened risk of cirrhosis [39,40]. A total of 4057 lncRNAs were identified in fibrotic NASH samples and normal hepatic samples [27]. Thus, lncRNAs are widely expressed in human liver fibrosis. Delineating lncRNA expression patterns will help to better recognize the function of lncRNAs in liver fibrosis.

Among those identified lncRNAs, some lncRNAs were firmly linked to numerous proteins and collagen genes that regulate the ECM during fibrotic scar formation, according to coexpression analyses [37]. Eight DE lncRNAs and three pairs of coexpressed lncRNAs-mRNAs were confirmed and enriched in growth factor binding terms, such as connective tissue growth factor (CTGF), *netrin-4* (*NTN4*), and *fibroblast growth factor 2* (*FGF2*), and the coexpressed lncRNAs may be involved in liver injury activation and human HSCs [38]. GO (gene ontology) and Kyoto Encyclopedia of Genes and Genomes (KEGG) analyses were enriched mainly in human HSC activation due to participation in growth factor binding, the DNA packaging/bending complex, and the Hippo signalling pathway [38]. Three lncRNAs were found to be closely related to *CTGF*, *FGF2*, and *NTN4* by lncRNA-mRNA coexpression analysis [38]. Furthermore, through experimental verification, 77 NASH-related target mRNAs were found to take part in 137 pathways, such as tumour necrosis factor (TNF) and TGFβ1 signalling, insulin resistance, and ECM maintenance [27]. NEAT1, HULC, and MALAT1 were more highly expressed in fibrotic tissues than in normal tissue [27]. These results show that lncRNAs are potentially important contributors to the progression and formation of liver fibrosis in humans.

### 3.2. LncRNAs Are Widely Expressed in Animal Models

A comprehensive lncRNA expression profiling study of mouse/rat liver fibrosis is very important to research the mechanism of human liver fibrosis and liver-related diseases. As classical model organisms, mice and rats are widely used in the study of human liver fibrosis.

DE lncRNAs-mRNAs were observed between activated and quiescent mouse HSCs. Twenty-four DE lncRNAs (12 upregulated and 12 downregulated) and 529 DE mRNAs (155 upregulated and 374 downregulated) were found to be dysregulated in activated mouse HSCs [41]. When comparing activated and quiescent HSCs, the lncRNAs NONRATT021402.2, NONRATT016788.2, and NONRATT012636.2 were upregulated, and the lncRNAs NONRATT024061.2, NONRATT019720.2, and NONRATT007863.2 were downregulated [41]. Furthermore, the upregulation of lncRNA NONRATT013819.2 and *lysyl oxidase* (*Lox*) mRNA in HSCs and fibrotic livers was associated with the ECM-related signalling pathway [41]. These results showing potentially vital functions for NONRATT013819.2 and *Lox* in ECM remodelling during HSC activation indicate that an abnormally regulated lncRNA-mRNA network might supply new hepatic fibrosis treatment measures.

Analysis of the ceRNA network between lncRNAs and miRNAs may help expand our understanding of the possible mechanism of hepatic fibrosis [20,36,42]. DE lncRNAs, miRNAs and mRNAs in mice were obtained from the National Center for Biotechnology Information (NCBI) database (miRNAs in GSE7727, lncRNAs and mRNAs in GSE80601) [43]. The results show that the ceRNA network included 220 lncRNA nodes, 24 miRNA nodes, 164 mRNA nodes and 1149 edges [43]. GO analysis of the lncRNAs based on the ceRNA network revealed that platelet-derived growth factor binding and cytokine and collagen regulation were significantly altered [43]. Correspondingly, KEGG pathway analysis indicated that the TGF-β, toll-like receptors (TLR), and peroxisome proliferator-activated receptor signalling pathways were significantly enriched. The above findings provide some new potential lncRNA markers for the clinical diagnosis and treatment of hepatic fibrosis.

The hepatotoxic substance carbon tetrachloride (CCl_4_) has been used extensively in studies related to liver fibrosis [44,45]. Determining DE lncRNAs and mRNAs in CCl_4_-induced rat liver fibrotic tissue by RNA-seq technology is important for understanding liver fibrosis. A total of 231 lncRNAs (102 upregulated and 129 downregulated) and 1036 mRNAs were DE in CCl_4_-treated fibrotic rat liver tissues compared with normal liver tissues [46]. It is worth noting that the KEGG pathways of the DE mRNAs that are essential for liver fibrosis development are enriched predominantly in ECM–receptor interaction, the phosphatidylinositol 3 kinase (PI3K)-AKT (protein kinase B) signalling pathway, and focal adhesion pathways [46]. These identified DE lncRNAs may serve as potential therapeutic targets and diagnostic biomarkers for liver fibrosis.

## 4. LncRNAs Are Involved in the Inhibition of Liver Fibrosis 

Several lncRNAs, including MEG3 [47], GAS5 [8], Gm5091 [19], NR_002155.1 [46], and HIF1A-AS1 [48], have been demonstrated to inhibit liver fibrosis (Table 1). These lncRNAs have similar characteristics, such as downregulated expression patterns in liver fibrosis, and they can inhibit HSC activation and suppress fibrosis progression upon overexpression. Therefore, enhancing the expression of these lncRNAs may provide a new mechanism to treat liver fibrosis.

### 4.1. MEG3

The lncRNA MEG3 is expressed in numerous normal tissues and acts as a tumour suppressor [29,72]. MEG3 expression levels were significantly decreased in both human fibrotic livers and CCl_4_-induced mouse liver fibrosis models [44]. Furthermore, MEG3 exerted an antifibrotic effect on TGF-β1-induced Lieming Xu-2 (LX-2) cell activation. MEG3 overexpression in TGF-β1-treated LX-2 cells could provoke p53 activation and cytochrome c release, subsequently mediating caspase-3-dependent apoptosis [44]. Chen et al. reported that MEG3 serum levels are low in chronic hepatitis B (CHB) patients and negatively correlate with the degree of liver fibrosis [47]. In addition, MEG3 has other mechanisms of inhibiting HSC activation [49]. MEG3 overexpression and its interaction with smoothened (SMO) participated in the suppression of epithelial to mesenchymal transition (EMT). On the other hand, miR-212 regulated the Hh pathway to mediate the effects of MEG3 on the EMT process [49]. These results suggest that MEG3 may play an important role in liver fibrosis progression and stellate cell activation and may serve as a novel potential diagnostic biomarker and therapeutic target for liver fibrosis.

### 4.2. GAS5

The lncRNA GAS5 is located at the prostate cancer-associated locus 1q25 [73], which comprises 12 exons and introns and encodes 10 box C/D small nucleolar RNAs (snoRNAs). As a tumour-suppressor gene [74,75], GAS5 can act as a sponge platform for miR-23a and decrease miR-23a expression levels and then increase phosphate and tension homology deleted on chromosome ten (PTEN); this increase further affects the downstream PI3K/Akt/mammalian target of rapamycin (mTOR)/Snail signalling pathway, increasing E-cadherin expression levels and α-SMA and decreasing collagen I expression levels in rat HSCs [8]. Moreover, by acting as a ceRNA for miR-222, GAS5 increased the p27 protein level, thus inhibiting the proliferation and activation of HSCs [50]. Consequently, the lncRNA GAS5/miR-23a/PTEN/PI3K/Akt/mTOR/Snail and GAS5/miR-222/p27 signalling pathways can provide molecular targets for the treatment of liver fibrosis.

### 4.3. Gm5091

Gm5091 (an intergenic lncRNA, 1179 bp) is located on Chr17 and was identified in a high-resolution anatomical atlas from a mouse embryo transcriptome [76]. Gm5091 was significantly downregulated during alcohol-induced hepatic fibrosis [19]. It has been demonstrated that Gm5091 plays an important role in reactive oxygen species (ROS) levels, negatively regulating cell migration, interleukin-1β (IL-1β) secretion, and the expression of collagen I and mouse HSC activation markers, including α-SMA and desmin. It is widely considered that miR-27, miR-24, and miR-23 can promote progression to liver fibrosis through HSC differentiation and proliferation by activating smad4 and TGF-β [77,78]. As a new regulator of alcoholic hepatic fibrosis progression, sponging of miR-27b/23b/24 by lncRNA Gm5091 alleviates mouse alcoholic hepatic fibrosis and thereby increases TGF-β levels [19]. Analysis of the mechanism of action of lncRNA Gm5091 could help us understand the process of alcoholic hepatic fibrosis.

### 4.4. NR_002155.1

The novel lncRNA NR_002155.1 was identified by RNA sequencing of CCl_4_-induced fibrotic rat liver tissue [46]. NR_002155.1 expression was downregulated in CCl_4_-treated liver samples compared to untreated liver samples. In addition, NR_002155.1 overexpression inhibited rat HSC-T6 cell proliferation according to cell counting kit-8 (CCK-8) assay results at 48 and 72 h [46]. Thus, NR_002155.1 may repress the proliferation of HSCs and then inhibit liver fibrosis caused by CCl_4_.

### 4.5. HIF1A-AS1

As a member of the ten-eleven translocation (TET) protein family, TET3 is closely associated with HSC activation and is reduced significantly in LX-2 HSCs activated by TGF-β1 [48]. LncRNA HIF1A-AS1 expression levels increased significantly in LX-2 cells treated with siRNA TET3 (siTET3) [48]. Furthermore, lncRNA HIF1A-AS1 silencing reduces apoptosis and promotes LX-2 cell proliferation [48]. These results illustrated that TET3 can mediate human HSC activation by modulating the expression of HIF1A-AS1 [48]. Thus, the TET3-lncRNA HIF1A regulatory axis will help us to understand the pathogenesis of liver fibrosis-related diseases.

## 5. LncRNAs Are Involved in the Promotion of Liver Fibrosis

Some lncRNAs are upregulated in liver fibrotic tissues and HSCs (Table 1), which promotes the progression of liver fibrosis through multiple functional mechanisms (for example, acting as ceRNAs and directly binding proteins) [60,79]. These upregulated lncRNAs may be biomarkers and treatment targets of liver fibrosis.

### 5.1. LncRNAs Are Involved in Liver Fibrosis by Binding to Proteins

#### 5.1.1. SCARNA10

The lncRNA ENSMUST00000158992 (named SCARNA10) is associated with the liver fibrosis stage and is increased in the liver tissues and serum of patients with progressive hepatic fibrosis, as well as in the liver tissues of mice with advanced liver fibrosis [51]. SCARNA10 is a novel positive regulator of TGF-β signalling in hepatic fibrogenesis, and can aggravate liver fibrosis mainly by promoting TGF-β signalling via inhibition of polycomb repressive complex 2 (PRC2) binding to the promoters of genes involved in the TGF-β pathway [51]. Specifically, SCARNA10 interrelates with PRC2 to regulate Smad2/3 and α-SMA expression; for example, SCARNA10 silencing is accompanied by decreases in TGFβ, TGFβRI, SMAD2, SMAD3, and kruppel-like factor 6 (KLF6) levels [51]. All these data suggest that SCARNA10 is not only a possible therapeutic target but also a potential diagnostic marker for liver fibrosis.

#### 5.1.2. Linc-SCRG1

Linc-SCRG1 (a lncRNA with a transcript length of 3118 bp) is expressed in only humans and specifically combines with increased tristetraprolin (TTP) protein levels during liver fibrosis progression in activated LX2 cells induced by TGF-β and in human tissues [52]. Linc-SCRG1 repressed TTP and restricted its degradation of the target genes TNF-α and matrix metallopeptidase-2 (MMP-2), thus inactivating the HSC phenotype [52]. This result suggests that linc-SCRG1 inhibition is a therapeutic approach to inactivate HSCs and inhibit human liver fibrosis.

#### 5.1.3. Lnc-LFAR1

Lnc-LFAR1 promoted mouse liver fibrosis [34]. Lnc-LFAR1 silencing decreases TGFβ-induced hepatocyte apoptosis in vitro, impairs HSC activation, and attenuates both CCl_4_- and bile duct ligation-induced liver fibrosis in mice [34]. In addition, direct binding of lnc-LFAR1 to Smad2/3 promotes its phosphorylation in the cytoplasm, thereby inducing TGF-β and Notch pathway activation [34]. The inhibition of HSC apoptosis and the inactivation of HSCs are currently recognized as mediating the resolution of liver fibrosis [3,34]. Therefore, lnc-LFAR1 may be a candidate target for fibrosis treatment.

### 5.2. LncRNAs Are Involved in Liver Fibrosis as CeRNAs that Bind to MiRNAs

#### 5.2.1. NEAT1

LncRNA NEAT1 may function as a miRNA sponge in hepatic diseases. NEAT1 is involved in the regulation of sorafenib resistance in hepatocellular carcinoma cells by sponging miR-335 expression [79] and accelerates liver fibrosis progression by regulating the miR-122- KLF6 axis [45]. In a recent article, Kong et al. found that NEAT1 may regulate miR-29b localization in the cytoplasm [20,59]. A previous study demonstrated that opposite expression patterns of NEAT1 (upregulation) and miR-29b (downregulation) were present during CCl_4_-induced liver fibrosis [45]. As an important regulatory molecule, insulin-like growth factor binding protein-related protein 1 (IGFBPrP1), interacts with TGF-β1 and promotes liver fibrosis [80,81]. Importantly, IGFBPrP1 increased the levels of autophagy-related protein 9A (Atg9a), NEAT1, and autophagy, whereas it decreased the level of miR-29b in mouse HSCs and liver tissues [20]. Importantly, IGFBPrP1-induced HSC autophagy and activation are stimulated and inhibited by NEAT1 and miR-29b, respectively [20]. Atg9a contributes to IGFBPrP1-induced HSC autophagy and activation. Thus, the NEAT1/miR-29b/Atg9a regulatory axis may offer a new understanding of the pathology and treatment of liver fibrosis. Otherwise, NEAT1 is a transcriptional target of p53 and modulates p53-induced transactivation and tumour-suppressor function [82,83]. Involvement of the p53 signalling pathway in the regulation of liver fibrosis has been demonstrated [54]. These results suggest that NEAT1 regulates liver fibrosis via p53.

#### 5.2.2. Hotair

The lncRNA Hotair located in the homeobox complex (HOXC) locus has been studied extensively in cancer, and the findings strongly suggest that Hotair promotes cancer progression [21,22,42,84]. Bian et al. found that Hotair was significantly upregulated in CCl_4_-induced human fibrotic livers, mouse liver fibrosis models, and HSCs activated by TGF-β1 stimulation [60]. Furthermore, Hotair directly bound to miR-148b and regulated the DNMT1(DNA (cytosine-5-)-methyltransferase 1)/MEG3/p53 pathway in HSCs. It is interesting to note that Hotair controls the repression of MEG3 by different pathways, which may be caused by the localization of Hotair in HSCs [60]. In another pathway, Hotair induced miR-29b downregulation and attenuated its control of epigenetic regulation, causing aggravated PTEN methylation, which promotes the progression of liver fibrosis in mice [59]. These data show that Hotair suppression may provide a potential therapeutic possibility for inhibiting liver fibrosis.

#### 5.2.3. HOTTIP

The lncRNA HOTTIP plays key roles in multiple human cancers, including colorectal cancer, hepatocellular carcinoma, and gastric cancer [28,85]. HOTTIP has been identified in human hepatocellular carcinoma (HCC) samples and is related to the disease outcomes and clinical progression of HCC patients [28]; in addition, HOTTIP is dysregulated in the early stage of human HCC [86]. HOTTIP is involved in regulating the progression of mouse liver fibrosis by promoting HSC activation as a ceRNA for miR-148a and miR-150 [35,61]. MiR-148a downregulation by HOTTIP plays a pivotal role in this pathogenic process [61]. Notably, high levels of HOTTIP downregulate miR-148a, increase the expression levels of the miR-148a targets TGF-beta receptor type-1 (TGFBR1) and TGF-beta receptor type-2 (TGFBR2) and thus contribute to liver fibrosis [61]. A later study showed that HOTTIP, as a ceRNA for miR-150, increases serum response factor (SRF) expression and induces mouse HSC activation [35]. This suggests the underlying mechanisms of HOTTIP in liver fibrosis and is predicted to provide new insights for therapeutic strategies.

#### 5.2.4. SNHG7

Small nuclear RNA host gene 7 (SNHG7) is upregulated in many cancer tissues and cell lines and may be a possible oncogene [87,88]. SNHG7 expression was significantly increased in human fibrotic liver tissues compared to control tissues [36]. Loss of SNHG7 induced the inhibition of mouse HSC activation and liver fibrosis in vivo, whereas miR-378a-3p downregulation blocked the effects of SNHG7 loss on mouse HSC activation. It has been demonstrated that miR-378a-induced dishevelled segment polarity protein 2 (DVL2) is responsible for activation of the Wnt (wingless/integrated)/b-catenin pathway by SNHG7 [36]. Understanding the SNHG7 functional mechanism in mouse HSCs is important for the analysis of liver fibrosis progression.

#### 5.2.5. PVT1

The lncRNA plasmacytoma variant translocation 1 (PVT1) is upregulated in activated HSCs and fibrotic liver tissues [62]. PVT1 depletion weakened collagen accumulation in vivo and inhibited HSC activation; it also reduced type I collagen and α-SMA levels and HSC proliferation in vitro [62]. Patched1 (PTCH1) is a negative regulator of the Hh pathway [63] and is enhanced by PVT1 knockdown. PVT1 epigenetically downregulates PTCH1 expression by competitively binding miR-152, thus promoting the EMT process in liver fibrosis [62]. Therefore, PVT1 acts through the EMT process, and the Hedgehog pathway regulates HSC activation.

#### 5.2.6. LncRNA-ATB

LncRNA activated by transforming growth factor beta (LncRNA-ATB) has been reported to be related to hepatocellular carcinoma invasion by regulating miR-200a target genes [89]. In another viral liver disease model, lncRNA-ATB was upregulated in hepatitis C virus-related fibrotic liver tissues and included the common binding sites for β-catenin and miR-200a [64]. In this study, downregulated lncRNA-ATB expression levels dysregulated endogenous miR-200a and then reduced β-catenin expression to suppress LX-2 cell activation [64]. Thus, the lncRNA-ATB/miR-200a/β-catenin regulatory axis likely plays important roles in the progression of liver fibrosis in hepatitis C virus (HCV) patients [64]. This suggests that lncRNA-ATB knockdown might be a potential therapeutic target for HCV-related liver fibrosis.

#### 5.2.7. MALAT1

The lncRNA MALAT1 is widely expressed and contributes to oncogenesis and tumour metastasis [90]. MALAT1 expression is upregulated and regulates sirtuin 1 (SIRT1) in liver fibrosis [91]. In addition, SIRT1 can stimulate the deacetylation of Smad3 (as a downstream mediator of the TGF-β signalling pathway) and then reduce Smad3 binding to fibrogenic gene promoters (collagen type I gene promoters) [83]. Thus, SIRT1 activation decreases TGF-β signalling and weakens TGF-β-stimulated collagen expression [92]. Moreover, MALAT1 affects the proliferation, cell cycle, and activation of primary mouse HSCs as a ceRNA regulating RAS-Rac1 (related C3 botulinum substrate 1) via miRNA-101b [65]. In another report, Dai et al. found that MALAT1 expression was upregulated in exosomes derived from arsenate-treated L-02 cells, as well as in LX-2 cells exposed to exosomes derived from arsenite-treated L-02 cells. Exosomal MALAT1 regulated COL1A2 by binding directly to miRNA-26b and thereby elevated LX-2 cell activation. These results suggest that exosomal MALAT1 is an important regulator in liver fibrosis induced by arsenite and highlight a mechanism of action for arsenite fibrogenesis [66]. In NASH, MALAT1 is highly expressed in fibrotic tissues relative to normal tissues [27]. Expression of MALAT1 and its target gene CXCL5 (C-X-C motif chemokine ligand 5) was upregulated in fibrotic livers and activated HSCs [27]. High MALAT1 expression may promote the progression of liver fibrosis in NASH patients via mechanisms relating to inflammatory chemokines [27]. These findings show that MALAT1 plays important roles in liver fibrosis, suggesting that it could be a prospective biomarker for liver fibrosis-related diseases.

### 5.3. Complicated Molecular Mechanisms of LncRNAs Involved in Liver Fibrosis

#### 5.3.1. LincRNA-p21

LincRNA-p21 (hepatocyte long intervening noncoding RNA-p21) is significantly upregulated during liver fibrosis [53] and was initially recognized as a transcriptional target of p53 that induces p53-dependent apoptosis in doxorubicin-treated mouse embryo fibroblasts [54]. LincRNA-p21 regulates liver fibrosis progression in a variety of ways as a ceRNA regulator. Tu et al. demonstrated that lincRNA-p21 strengthens TGF-β signalling by acting as a downstream effector of TGF-β signalling and promotes mouse liver fibrosis by interacting with miR-30 [53]. In contrast, hepatocyte lincRNA-p21 knockdown mediated CCl_4_-induced inflammation and liver fibrosis, and ectopic miR-30 expression in hepatocytes yielded similar results [53]. On the other hand, lincRNA-p21 inhibition of the Wnt/β-catenin pathway is related to the effects of salvianolic acid B on HSC activation [55]. Moreover, lincRNA-p21 inhibits HSC activation, at least in part, through the Wnt/β-catenin pathway mediated by miR-17-5p [55]. Yu et al. reported that lincRNA-p21 strengthened PTEN expression via competitively binding miR-181b [56]. In addition, lincRNA-p21 suppressed the cell cycle proliferation and progression of primary HSCs by enriching p21 expression [57]. Furthermore, compared with those in healthy patients, serum lincRNA-p21 levels were significantly downregulated in liver cirrhosis patients, particularly those with decompensation [57], and might be a possible biomarker of liver fibrosis in chronic hepatitis B patients [58]. The important roles of hepatocyte lincRNA-p21 suggest that it serves as a potential target for liver fibrosis therapy.

#### 5.3.2. HULC

The lncRNA highly upregulated in liver cancer (HULC) is closely related to liver diseases [93,94], which play a key role in liver fibrosis [67]. HULC expression levels were increased in liver tissues from nonalcoholic fatty liver disease rats [67]. p38 mitogen-activated protein kinase (MAPK) and c-Jun N-terminal kinase (JNK) mRNA expression in liver tissue from NAFLD rats was decreased by treatment with HULC small interfering RNA (siRNA). Thus, HULC inhibition enhanced the degree of hepatic fibrosis and reduced hepatocyte apoptosis by suppressing the MAPK signalling pathway in rats with non-alcoholic fatty liver disease [67]. Despite these findings, the specific target gene of HULC in the MAPK signalling pathway remains unclear. However, this should not prevent lncRNA HULC from being considered a potential hepatic fibrosis target in non-alcoholic fatty liver disease.

## 6. Both Inhibition and Promotion of Liver Fibrosis by LncRNA

LncRNA H19 (both hepatic and serum exosomal long noncoding RNA H19) is an imprinted and maternally expressed gene that is conserved between humans and mice and is involved in the regulation of cell proliferation and differentiation [95,96]. A previous study found that H19 is upregulated in CLI animal models and human liver diseases [92,97,98]. H19 plays an important role in the process of liver fibrosis, with inhibition and promotion mediated by different targets.

### 6.1. Inhibition of Liver Fibrosis by H19

Aberrant DNA methylation could lead to monoallelic to biallelic changes in H19 expression in hepatocellular carcinoma and play a potential role in cell activation and proliferation [69,99]. In the regulation of liver fibrosis, H19 has a relatively complex mechanism of action. H19 is an underlying regulator of IGF1R in HSCs [69]. IGF1R and methyl-CpG-binding protein 2 (MeCP2) were significantly upregulated in HSCs and fibrotic tissues, and the opposite pattern was detected for H19 [69]. MeCP2 knockdown inhibits HSC proliferation and increases H19 expression in activated HSCs [69]. H19 overexpression may suppress IGF1R expression and HSC activation [69]. MeCP2 targeted the protein IGF1R and thereby negatively regulated H19 [69]. The MeCP2/H19 axis has potential importance in the progression of liver fibrosis and contains possible targets for future therapeutic intervention.

The mechanism of action of H19 in liver fibrosis is related to DNA methylation, and the DNA methylation reader protein DNMT1 regulates the lncRNA H19/ERK (extracellular regulated protein kinases) signalling pathway in HSC activation and fibrosis [68]. Reduced lncRNA H19 expression and increased DNMT1 expression and lncRNA H19 promoter methylation were found in activated HSCs and fibrotic rat liver tissue [68]. In addition, DNMT1 reduction and overexpression resulted in the opposite expression pattern of H19 in activated HSCs [68]. Furthermore, the expression of p-ERK1/2 was increased by H19-siRNA in HSCs [68]. Therefore, the DNMT1-lncRNA H19 epigenetic pathway plays important roles in liver fibrosis.

### 6.2. Promotion of Liver Fibrosis by H19

H19 is involved in the control of cholestatic liver fibrosis. E-Box-binding homeobox 1 (ZEB1) represses the promoter activity and gene transcription of epithelial cell adhesion molecule (EpCAM). Hepatic H19 RNA activation promoted cholestatic liver fibrosis in mice by binding to ZEB1 and thereby increasing EpCAM expression [70]. Thus, H19 could be a profibrotic mediator of cholestatic live fibrosis and may highlight a direction for the improvement of therapeutic strategies targeting lncRNAs [70]. In biliary atresia (BA)-related liver fibrosis, the severity of fibrotic liver injuries in BA patients is correlated with hepatic exosomal H19 levels [71]. H19 plays a key role in cholestatic liver injury and cholangiocyte proliferation in BA patients by regulating the sphingosine 1-phosphate receptor 2 (S1PR2)/sphingosine kinase 2 (SphK2) and let-7/high-mobility group AT-hook 2 (HMGA2) axis [71]. On the other hand, cholangiocyte-derived exosomal H19 plays an important role in the development of cholestatic liver fibrosis by stimulating HSC activation and differentiation and represents a possible therapeutic target and diagnostic biomarker for cholangiopathies [100]. This suggests that H19 may act as a noninvasive potential therapeutic target and diagnostic biomarker for cholestatic liver fibrosis.

## 7. Signalling Pathways Involved in Liver Fibrosis

To date, lncRNAs involved in multiple signalling pathways, such as the TGF-β, AKT/mTOR/p27, Hh, SIRT/P53, MAPK, and Wnt/β-catenin signalling pathways, have been demonstrated to regulate liver fibrosis [12,95] (Figure 3). These signalling pathways may also become important therapeutic targets for liver fibrosis.

The interaction between lncRNAs and the TGF-β signalling pathway is the most well known in liver fibrosis. As is a master profibrogenic cytokine, TGF-β controls liver physiology and pathology during the initial process of liver injury–inflammation–fibrosis [92,96,97]. TGF-β influences the different biological processes of liver fibrogenesis, including the circadian rhythm, epigenetics, reactive oxygen species generation, metabolism, senescence, EMT, and endothelial to mesenchymal transition [97]. To date, it has been found that lncRNAs cooperate with the TGF-β signalling pathway to regulate HSC activation and hepatic fibrosis. Inhibition of lncRNAs (HIF1A-AS [48], Gm5091 [77,78], and lincRNA-p21 [53]) and overexpression of lncRNAs (MALAT1 [92], SCARNA10 [51], Lnc-LFAR1 [34], and HOTTIP [61]) involved in the TGF-β signalling pathway could promote HSC activation and subsequently induce hepatic fibrosis. Thus, lncRNAs can reveal the mechanism by which the TGF-β signalling pathway is involved in regulating liver fibrosis based on noncoding RNAs. These lncRNAs and TGF-β are promising targets for treating fibrosis.

Another important pathway is the AKT/mTOR/p27 signalling pathway, which impacts downstream pathways, such as the P53 signalling pathway, the MAPK signalling pathway, and the Hh signalling pathway. LncRNAs are involved in regulating the activity of the above pathways (Figure 3). Specifically, HOTAIR can regulate the AKT/mTOR/p27 and P53 signalling pathways via PTEN methylation and MEG3 methylation, respectively [59,60]. Furthermore, GAS5 [8,50] and lincRNA-p21 [56] regulate the AKT/mTOR/p27 signalling pathway as ceRNA-binding miRNAs. On the other hand, the Wnt signalling pathway plays important roles in liver fibrosis. LincRNA-p21 [56] and ATB [64] can competitively combine with miRNA to change the protein expression level of β-catenin.

In sum, lncRNAs provide new perspectives and knowledge to reveal and understand the mechanisms of liver fibrosis and are potential new targets for inhibiting fibrosis.

## 8. Conclusions and Future Perspectives

LncRNAs have emerged as important regulators of liver fibrosis, and abnormal lncRNA expression has been linked to HSC activation. The regulatory network of lncRNAs in liver fibrosis is believed to be very complicated, and the underlying molecular mechanisms by which lncRNAs cause liver fibrosis remain largely ambiguous. A lncRNA transcriptome database of liver fibrosis data from humans and mice/rats has been constructed and provides the ability to select biomarkers and targets. DE lncRNAs between normal and hepatic fibrotic tissues not only suggest that lncRNAs may be involved in hepatic fibrosis progression but also imply that lncRNAs may be biomarkers for the clinical diagnosis of hepatic fibrosis. Thus, we hope that this review increases awareness of the dominant roles of many lncRNAs and their multifaceted regulatory linkages in the progression of liver fibrosis. More attention should be paid to screening more effective lncRNAs for the diagnosis and treatment of liver fibrosis and to developing effective lncRNAs for clinical use.

LncRNAs have complicated structures, such as their secondary structures, which determine multiple mechanisms of action [101,102,103]. There are two main models of the mechanisms by which lncRNAs are involved in regulating liver fibrosis: by acting as a ceRNA and a direct binding protein. There may be additional mechanisms of action for lncRNAs in liver fibrosis. It is well known that circulating RNAs (e.g., lncRNAs) are stable and that their levels can be easily quantified by quantitative RT-PCR or high-throughput assays [104]. To date, noncoding RNAs have been proven to more accurately predict liver fibrosis than the conventional platelet count, prothrombin international normalized ratio (INR), or albumin level [105]. Therefore, it is speculated that lncRNAs have great potential in the diagnosis and treatment of liver fibrosis. Depletion of upregulated lncRNAs or overexpression of downregulated lncRNAs in patients are the main mechanisms for establishing lncRNA-based therapy. Strategies to modulate lncRNA expression in different liver diseases, including fibrosis, have been successfully implemented in preclinical models. Although these therapeutic strategies for manipulating lncRNA levels appear to have promising clinical applications, their safety and reliability remain substantial challenges. The development of improved knowledge and advanced technology will help us apply lncRNA-directed therapy to suppress liver fibrosis and hepatic diseases in the future.

## Figures and Tables

**Figure 1 ijms-21-01482-f001:**
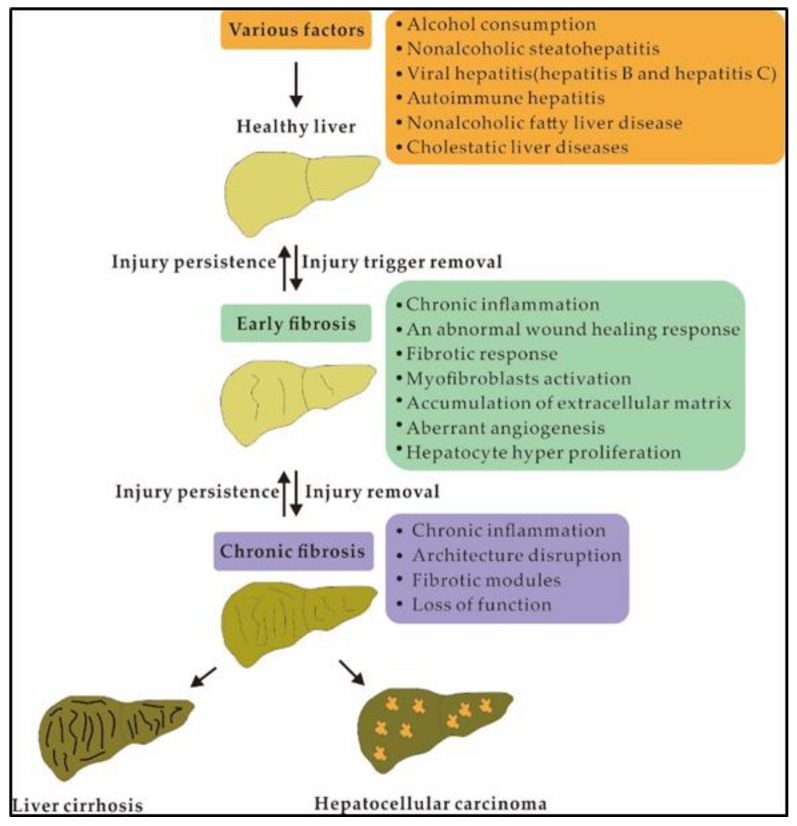
Pathogenesis of liver fibrosis.

**Figure 2 ijms-21-01482-f002:**
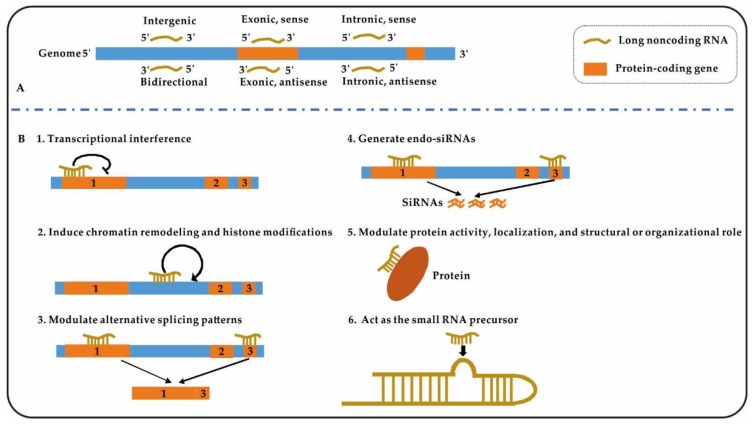
Classification and mechanisms of action of long noncoding RNAs (lncRNAs). (**A**), Classification of lncRNAs according to protein-coding genes; (**B**), Six mechanisms of “lncRNAs” in Figure 2.

**Figure 3 ijms-21-01482-f003:**
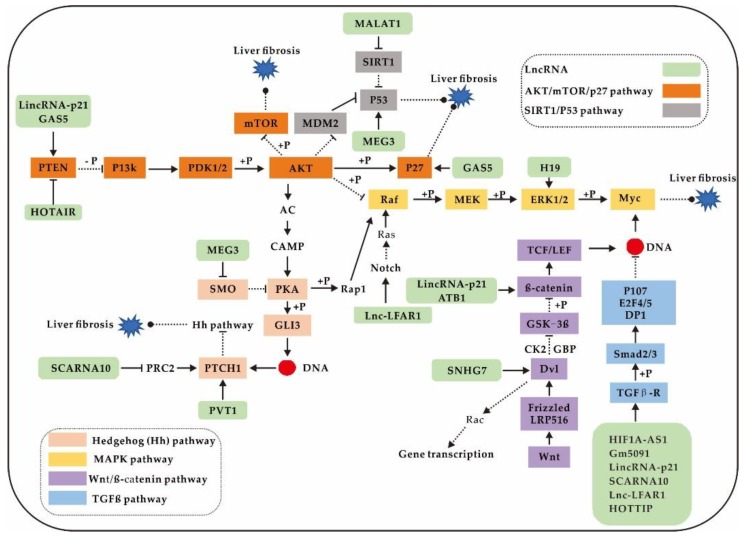
LncRNAs are involved in the regulation of liver fibrosis by multiple signalling pathways.

**Table 1 ijms-21-01482-t001:** LncRNAs are involved in regulating liver fibrosis.

LncRNAs	Targets	Functions	Ref.
**Inhibition of liver fibrosis**			
MEG3	SMO	MEG3 overexpression is involved in the suppression of HSC activation	[49]
GAS5	MiR-23a, miR-222	GAS5 can competitively decrease miR-23a and miR-222 expression levels, which respectively increase PTEN and p27 levels, to further decrease HSCs	[8,50]
Gm5091	MiR-27b, miR-23b, miR-24	Gm5091 increases TGF-β by sponging miR-27b/23b/24 to alleviate mouse alcoholic hepatic fibrosis	[19]
NR_002155.1	-	NR_002155.1 overexpression inhibits rat HSC-T6 cell proliferation	[46]
HIF1A-AS1	TET3	HIF1A-AS1 overexpression inhibits LX-2 cell proliferation	[48]
**Promotion of liver fibrosis**			
SCARNA10	PRC2	SCARNA10 interacts with PRC2 to regulate the expression of α-SMA and Smad2/3	[51]
Linc-SCRG1	Tristetraprolin (TTP)	Linc-SCRG1 represses TTP and restricts its degradation of the target genes TNF-a and MMP-2; it also represses the TTP-induced inactivating effect on the HSC phenotype	[52]
Lnc-LFAR1	Smad2/3	Lnc-LFAR1 binds directly to Smad2/3 and promotes its phosphorylation, thereby promoting TGF-β and Notch pathway activation to accelerate liver fibrosis	[34]
NONRATT013819.2	*Lox*	LncRNA NONRATT013819.2-*Lox* network is associated with ECM remodelling during HSC activation	[41]
LincRNA-p21	P21, miR-181b, miR-30, miR-17-5p	Hepatocyte lincRNA-p21 greatly promotes CCl_4_-induced liver fibrosis and HSC activation	[53,54,55,56,57,58]
NEAT1	MiR-122, miR-29b	NEAT1 accelerates the progression of liver fibrosis by regulating miR-122/KLF6 and miR-29b/Atg9a	[20,45]
Hotair	MiR-148b, miR-29b	Hotair promotes liver fibrosis as an endogenous ‘sponge’ of miR-148b and miR-29b, which respectively regulate DNMT1/MEG3/p53 pathway expression and PTEN methylation in liver fibrosis	[59,60]
HOTTIP	MiR-148a, miR-150	HOTTIP promotes HSC activation as a ceRNA for miR-148a and miR-150	[35,61]
SNHG7	MiR-378a-3p	Loss of SNHG7 suppresses mouse HSC activation and liver fibrosis in vivo	[36]
PVT1	MiR-152	PVT1 epigenetically downregulates PTCH1 expression by competitively binding miR-152; PVT1 depletion attenuates collagen deposits in vivo and inhibits HSC activation	[62,63]
LncRNA-ATB	MiR-200a	LncRNA-ATB downregulates β-catenin expression by upregulating endogenous miR-200a to suppress LX-2 cell activation	[64]
MALAT1	MiRNA-101b, miRNA-26b	MALAT1 influences the proliferation, cell cycle, and activation of primary HSCs as a ceRNA	[27,65,66]
HULC	---	LncRNA HULC inhibition improves hepatic fibrosis and hepatocyte apoptosis by inhibiting the MAPK signalling pathway in rats with non-alcoholic fatty liver disease	[67]
**Both inhibition and promotion of liver fibrosis**			
H19	MeCP2, ERK1/2	H19 overexpression may suppress the expression of MeCP2/IGF1R and p-ERK1/2 to inhibit HSC activation	[68,69]
	ZEB1, let7	Hepatic H19 RNA activation promotes cholestatic liver fibrosis and biliary atresia (BA)-related liver fibrosis	[70,71]

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
