# Peer review of "The Roles and Mechanisms of lncRNAs in Liver Fibrosis"

_ijms, 2020, doi:10.3390/ijms21041482_

Round 1
Reviewer 1 Report
This review is well described and seems to be beneficial for researchers involved in this area.
Although the authors described lincRNA-p21 is a transcriptional target of p53, recent studies revealed that NEAT1 is also transcriptional target of p53. (Idogawa, et al. Int J Cancer 2017, Adriaens, et al. Nat Med 2016). The authors should cite these papers and reviewed related matters.
Author Response
Point-by-point responses to reviewers 1 comments/suggestions Dear reviewer, Thank you very much for reviewing our manuscript. We appreciate that the reviewer’s comments. All modifications in the revised manuscript are marked in red color. The followings are our point-by-point responses: Point 1: Although the authors described lincRNA-p21 is a transcriptional target of p53. (Idogawa, et al. Int J Cancer 2017, Adriaens, et al. Nat Med 2016). The authors should cite these papers and reviewed related matters. Response: Thanks for reviewer’s the favorable suggestions. More information and references have been added in revised manuscript (line 243-line 246, page 19). The p53 signalling pathway involved in regulation of liver fibrosis had been demonstrated. Then, recent studies revealed that NEAT1 is a transcriptional target of p53 and modulates p53-induced transactivation and tumor-suppressor function (Adriaens, et al., 2016; Idogawa, et al., 2017). To date, no direct evidence shows NEAT1 involved in regulation of liver fibrosis by p53. Maybe we could speculate about that.Reviewer 2 Report
The information embedded in this review is sound and the subject is of interest. there are a few concerns:
The English needs to be improved, to start with the first sentence of the abstract and the first sentence of the introduction.
Are lncRNAs sufficiently conserved over species that information for mice and rats are also true for humans? For example the mentioned NONRAT lncRNAs are they also found in humans? Make sure this information is clear and added to the table.
In figure 2 some lncRNA have a stimulating and some an inhibiting effect, it would be good to show the effect. So make stimulating and inhibiting arrows for the effects. Or make different figures?
H19 has an effect on both sides both inhibiting and stimulating, probably depending on the different targets, but that is very confusing now. That is probably due to the structure of the paper: It is too much a list of all the lncRNAs involved. For a good review it should be more comprehensive: describe the different processes in fibrosis and explain the role of the lncRNAs in these processes for example.
Author Response
Point-by-point responses to reviewers 2 comments/suggestions
Dear reviewer,
Thank you very much for reviewing our manuscript. We appreciate that the reviewer’s comments. All modifications in the revised manuscript are marked in red color. The followings are our point-by-point responses:
Point 1: The English needs to be improved.
Response: Thanks for reviewer’s suggestions. This manuscript proofreading has been done by a professional English speaking with science background at American Journal Experts Corporation (57D8-7E52-226B-08ED-BA1P).
Point 2: Are lncRNAs sufficiently conserved over species that information for mice and rats are also true for humans? For example the mentioned NONRAT lncRNAs are they also found in humans? Make sure this information is clear and added to the table.
Response: Despite the primary sequences of lncRNAs being weakly conserved across species, this does not imbue a lack of function (Li et al., 2019). The RNA structure of lncRNAs appears to be the main functional unit and evolutionary constraint (Johnsson et al., 2014). The certain conserved secondary structures originated from the structural motif might be crucial for the function of lncRNAs. Thus the function and action mechanism of lncRNA in mice and rats could provide the efficient reference for humans. Of seven NONRAT lncRNAs, NONRATT016788.2, NONRATT024061.2, NONRATT019720.2, NONRATT007863.2, and NONRATT013819.2 have the homologous genes with human. Interestingly the homologous gene of NONRATT013819.2 is Lox mRNA from mouse and human. Furthermore, NONRATT013819.2 has been added to the table 1.
Point 3: In figure 2 some lncRNA have a stimulating and some an inhibiting effect, it would be good to show the effect. So make stimulating and inhibiting arrows for the effects. Or make different figures?
Response: In order to conveniently view lncRNAs involved in regulation of different signalling pathways on the whole, Figure 2 has been modified according to the reviewer¢s suggestion and revised as Figure 3.
Point 4: H19 has an effect on both sides both inhibiting and stimulating, probably depending on the different targets, but that is very confusing now. That is probably due to the structure of the paper: It is too much a list of all the lncRNAs involved. For a good review it should be more comprehensive: describe the different processes in fibrosis and explain the role of the lncRNAs in these processes for example.
Response: Thanks for reviewer’s constructive comments. Karsdal et al. (2020), Rikhi et al.(2020), Campana et al. (2017) and Aydın et al. (2018) have been described the different processes of fibrosis in their reviews. Therefore, we did not introduce the different processes of fibrosis in a separate section, and elaborated it in ‘Introduction’ part. New figure (figure 1) has been provided in revised manuscript for readers to better understand the different processes of fibrosis in detail.
Liver fibrosis is the complex pathological process (Aydın and Akçalı, 2018). Different cell types activate myofibroblasts depending on the etiology of liver fibrosis (Iwaisako et al., 2014). As we know, lncRNAs take part in regulation pathogenesis of liver fibrosis. Some lncRNAs play a role even through all pathological period, such as SCARNA10 (Zhang et al., 2019). In revised manuscript, lncRNAs have been classified three parts according to their function in liver fibrosis, including “4. LncRNAs Involved in Liver Fibrosis Inhibition”, “5. LncRNAs Are Involved in the Promotion of Liver Fibrosis”, and “6. Both Inhibition and Promotion Liver Fibrosis by LncRNA”. Furthermore, the activate mechanism of H19 in liver fibrosis was described in “6. Both Inhibition and Promotion Liver Fibrosis by LncRNA” part.

Round 2
Reviewer 2 Report
The paper has been sufficiently improved!